# Integrative Innovation in Genioplasty: Advanced 3D Plate Design: Promoting Stability, Aesthetics, and Harmony Excellence

**DOI:** 10.3390/cmtr18030042

**Published:** 2025-09-22

**Authors:** Bruno Nifossi Prado, Lucas Cavalieri Pereira, Bianca Pulino, Raphael Capelli Guerra

**Affiliations:** 1Hospital Sírio-Libanês, Instituto de Ensino e Pesquisa, São Paulo 01308-050, Brazil; brunoprado8@gmail.com; 2Oral & Maxillofacial Surgery, São Leopoldo Mandic University, Campinas 13045-755, Brazil; dr.lucasmaxilofacial@hotmail.com; 3Department of Oral and Maxillofacial Surgery, Hospital Israelita Albert Einstein, São Paulo 05652-900, Brazil; 4Faculdade Israelita de Ciências da Saúde Albert Einstein, São Paulo 05652-900, Brazil

**Keywords:** genioplasty, chin osteotomy, finite element analysis, anatomical chin plate, facial aesthetics, plate fixation, mandibular surgery

## Abstract

Background: Genioplasty is a well-established surgical technique for reshaping the chin and enhancing facial harmony. However, conventional fixation methods may present biomechanical and aesthetic limitations. Objective: This study introduces and evaluates a novel Anatomical Chin Plate (ACP), designed to enhance mechanical performance and facial aesthetics compared to the conventional chin plate (CP). Methods: A three-dimensional finite element analysis (FEA) was conducted to compare stress distribution in ACP and CP models under a standardized oblique load of 60 N, simulating muscle forces from the mentalis and digastric muscles. Plates were modeled using Blender and analyzed using ANSYS software 2025 r2. Mechanical behavior was assessed based on von Mises stress, concentration sites, and potential for plastic deformation or fatigue failure. Results: The ACP demonstrated a significantly lower maximum von Mises stress (77.19 MPa) compared to the CP (398.48 MPa). Stress distribution in the ACP was homogeneous, particularly around the lateral fixation holes, while the CP exhibited concentrated stress between central screw holes. These findings indicate that the anatomical geometry of the ACP enhances load dispersion, reduces critical stress concentrations, and minimizes fatigue risk. Conclusions: The ACP design offers superior biomechanical behavior and improved aesthetic potential for genioplasty procedures. Its optimized shape allows for better integration with facial anatomy while providing stable fixation. Further studies are recommended to validate in vitro performance and explore clinical applicability in advanced genioplasty and complex osteotomies.

## 1. Introduction

The aesthetics of the face have attracted patients since the dawn of humanity. Several techniques have been developed to achieve facial harmony and symmetry. Jaw advancement surgeries, fillers, and genioplasty have been employed in the lower third of the face with excellent outcomes and clear indications [1].

Genioplasty is a well-established surgical procedure. Performed through osteotomies in the mental region of the mandible, this procedure involves separating the bone stumps and repositioning them in the necessary position, followed by a rigid fixation using plates and screws. This technique allows for advancement, setback, widening, narrowing, and increase or decrease in facial height [2].

An alternative to genioplasty is the placement of anatomical facial prostheses, which increase the volume and contour of the chin region without requiring osteotomies [3].

Finite element analysis (FEA) is commonly performed before in vitro and in vivo studies. In these studies, we can virtually simulate situations where forces are applied in different regions of the plates and evaluate the maximum stress and fatigue of the material [4]. This technique helps guide future studies and assess the feasibility of the product [5].

To revolutionize and improve this surgical approach, we developed an anatomical chin plate that enables multidirectional movement of the bone stumps, offering enhanced contour and anatomy to the facial aesthetic region. In this study, we compared a conventional chin plate (CP) with the new anatomical chin plate (ACP) using finite elements.

## 2. Materials and Methods

### 2.1. New Design

In genioplasty procedures, alterations in chin position and morphology are evident; however, deficiencies in the mentolabial region frequently demand adjunctive interventions to achieve optimal facial balance. The ACP was specifically developed to overcome this limitation by providing superior biomechanical performance, including improved load distribution, resistance to torsional and shear stresses, and reduction in stress shielding. These features contribute to enhanced fixation stability, controlled osteotomized segment displacement, and greater long-term predictability of mandibular biomechanics.

The ACP arose from the need to bring a greater aesthetic result to genioplasty, and bone advancement or setback is already well-established in both the literature and clinical practice. The aesthetic complement of the mentolabial sulcus in several patients often requires bone or fat grafts to obtain the desired aesthetic effect.

This new customized plate allows for precise planning of bone advancement or setback while maintaining the aesthetics of the mentolabial fold, which can be seen both in the profile and frontal analysis of the patient (Figure 1). The ACP It was projected and manufactured by EVOLVE^®^, São Caetano, Brazil.

To ensure the ACP fits accurately and achieves the planned aesthetic outcome, a guided osteotomy must be performed. A surgical guide is fabricated with high precision to match the planned osteotomy. This guide must be stabilized at both the top (occlusion of the lower teeth) and bottom using a screw (Figure 2). The screw used to stabilize the guide also serves as a drilling guide for the placement of the ACP.

During planning the ACP, the required size of the screws and their relationship with the dental roots were carefully determined to avoid future complications (Figure 3).

### 2.2. Modeling and Mechanical Analysis

The plates used in this study were modeled using Blender software version 3.x (Blender Foundation, 2025, Amstedam, The Netherlands). Titanium Ti-6AI-4V plates were used, with an elastic modulus of 110 GPa, Poisson’s coefficient of 0.34, and a yield strength of 880 MPa. These three-dimensional models were imported into the ANSYS (Mechanical Finite Element Analysis software 2025 r2) (© 2024 ANSYS, Inc., Canonsburg, PA, USA).

To simulate muscle loading in the chin region, an oblique force vector with a modulus of 60 N was applied, oriented at 45° in the sagittal plane, directed inferoposteriorly. This direction corresponds to the muscular forces exerted primarily by the mental muscles and the anterior belly of the digastric during functional contraction.

Based on the functional anatomy of the mentalis and anterior belly of the digastric muscles, the resulting vector of muscle force acting on the chin can be represented by an oblique vector inclined at 45°, directed inferoposteriorly (i.e., downward and backward).

If we consider a total net force of 60 N, the vector at 45° has equal components in X and Y, as follows:Fx = Fy = F⋅cos (45°) = 60⋅22 ≈ 42.4 NFx = Fy = F⋅cos (45°) = 60⋅22 ≈ 42.4 N

To evaluate the mechanical strength of the titanium plates used in genioplasty, three-dimensional finite element analysis was performed using the von Mises equivalent stress criterion. This criterion was selected because it is widely accepted in the flow prediction of ductile materials such as Ti-6Al-4V.

The plate model was subjected to a vector force of 60 N applied at 45° in the sagittal plane, representing the combined traction of the mentalis muscles and anterior belly of the digastric. Boundary conditions were applied to simulate rigid fixation of the plate to the mandibular blocks using screws that were restricted to all translational degrees of freedom.

The mesh was composed of second-order tetrahedral elements, and the solution was applied until the numerical residuals converged. Von Mises stresses were calculated throughout the body of the plate to identify the regions of stress concentration. Results were expressed in megapascals (MPa) and presented on a continuous chromatic scale.

## 3. Results

Two different configurations of titanium plates for genioplasty were analyzed with respect to the distribution of von Mises stresses under the same 60 N loading applied at 45°, simulating the combined vector action of the mentalis and anterior digastricus muscles. All models and their characteristics are summarized in Table 1.

For the conventional plate (Figure 4), the maximum recorded stress was 398.48 MPa, which was in the central region of the plate between the fixing holes. The stress distribution was concentrated in the geometric transition areas and near the orifices, suggesting a possible failure starting point under the cyclic loading. Despite the concentration, the values remained below the yield strength of grade 5 titanium (880 MPa), indicating structural safety under an isolated loading.

In contrast, for the ACP (Figure 5), the maximum stress was 77.19 MPa, distributed around the fixation holes on the lateral portions of the implant. The stress distribution was more homogeneous, with a significant reduction in peak concentrations, particularly in the critical transition regions between the holes and edges. The results suggest that anatomical geometry favors a more efficient dispersion of the load, contributing to greater resistance to fatigue and a reduced risk of local failure.

## 4. Discussion

Genioplasty is a surgical procedure that modifies the size, shape, and position of the chin in all directions [2]. When we combined genioplasty with internal fixation, regardless of the fixation type, the procedure proved to be stable regardless of movement (advance or retreat) when evaluating recurrence in hard and soft tissues [6].

In terms of aesthetics, genioplasty, when compared to chin implants, provided more satisfaction in patients undergoing osteotomy and produced fewer complications [3]. When we think about the osteotomy itself, a modification that has great acceptance is the Chin Wing [7]. Because of its great extension in most cases, it becomes necessary to have extra fixations, which would be an excellent indication for the ACP due to its high degree of resistance. In cases of minimally invasive osteotomies with different shapes [8], the ACP can be used to improve the quality of the soft tissue in the submental region.

We used finite elements as an accurate and cohesive method for evaluating the strength of a plate. It is known that the width, rather than the thickness, of a plate can influence the resistance, making the facts decisive for reducing the maximum stress under bending forces [9]. This is in accordance with the shape of our ACP, which, because it is extended, has 11 fewer tensions than a conventional plate in addition to delivering the necessary aesthetics for the region.

When evaluating the stability of genioplasty, one of the main concerns is the stability of the bone stump in relation to the musculature inserted in the chin. This fixation method is effective in preventing recurrence [5]. Initially, genioplasties were fixed with two bicortical screws; however, FEA has shown that a centrally located conventional plate produced more stability than two bicortical screws [10]. Thus, the placement of a plate, in addition to being more stable, is an easier procedure, particularly in cases of indentation and chin asymmetry.

The ACP was slightly thicker than the conventional plate. Plate thickness can be an important factor influencing the yield threshold and plate deformation. Further studies using plates of different thicknesses should be conducted in the future.

Among the various osteotomy techniques, the T-osteotomy, which removes the central part of the chin, reduces the width, and provides a new contour, is considered one of the least stable and requires fixation with greater stability. A finite element study found that the optimal stability was achieved with either a 1.2 mm plate and five screws or a 0.9 mm plate and seven screws [11]. This suggests that the new ACP, in addition to maintaining a thickness of 1.2 mm, also has six screws, which should be more than ideal to ensure perfect resistance to variations in T-osteotomy.

In advanced genioplasties, customized plates have demonstrated better distribution of strain stress and improved stabilization between the stem and segment [12]. Another benefit of customized plates is the incorporation of nanoparticles such as graphene into their composition; these alloys have been shown to reduce the stress distribution between 5% and 25% in the critical zones of titanium plates [13]. One disadvantage of this customization is that ACP will always depend on centers that work with proper software and a local company that customizes the ACP.

The thickness of the plate is a primary factor for stress distribution on the plates, promoting more resistance in advance [14,15]. The conventional plate has a thickness of 0.8 mm, whereas the customized ACP, at its smallest point, maintains a thickness of 1.2 mm, making it substantially more resistant to titanium fatigue. In addition to the thickness of the ACP board, a crucial change that can be a factor of more resistance can be its geometric shape. No other plate was designed to reproduce the anatomical form, focusing only on fixing the bone stumps letting the bone itself assume the aesthetic role.

## 5. Conclusions

The results indicate that the ACP model exhibits superior mechanical performance compared to the conventional CP, with significantly lower stress values and improved load distribution throughout the structure. These findings suggest a more favorable biomechanical behavior, particularly under cyclic or prolonged loading conditions, such as those associated with functional activities of the chin.

Further long-term studies are warranted to validate the effectiveness of the ACP, particularly through advanced biomechanical models and loading simulations, in order to assess the in vitro resistance of this new design. Such investigations may also extend to different types of osteotomies applied in genioplasty.

With validation and eventual clinical adoption, anatomical plates for genioplasty may enable the integration of osteotomy stability with prosthetic contouring, providing a unified technological approach to optimize both biomechanical reliability and facial esthetics.

## Figures and Tables

**Figure 1 cmtr-18-00042-f001:**
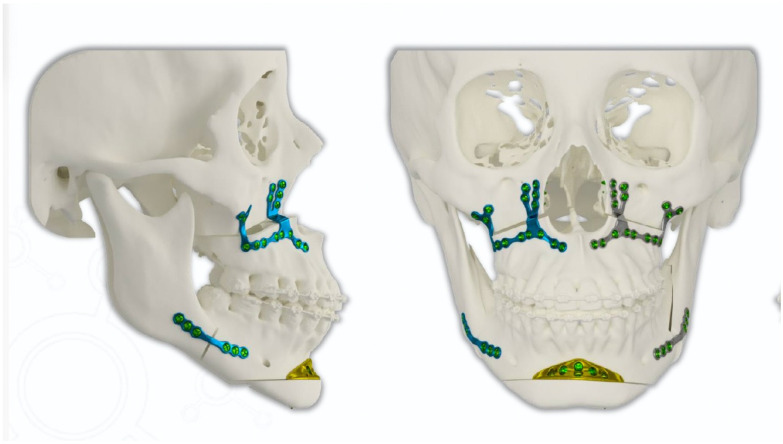
Anatomical Chin Plate (ACP) customized already, providing for the necessary bone advancement and volumization of the labial chin groove.

**Figure 2 cmtr-18-00042-f002:**
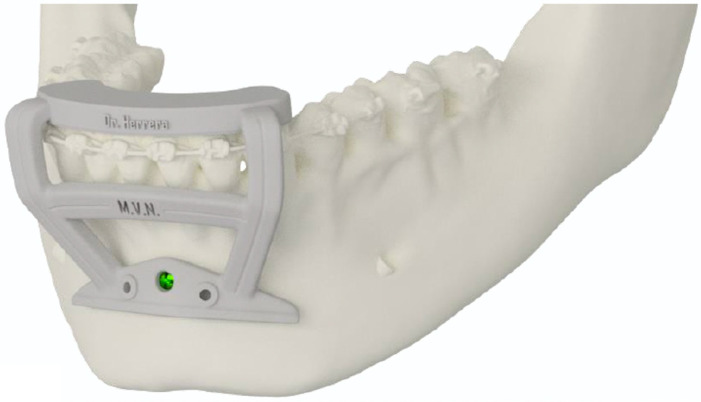
Cutting guide used to perform the correct osteotomy on the chin. This surgical guide has an upper support on the dental elements and a centralized screw at the bottom.

**Figure 3 cmtr-18-00042-f003:**
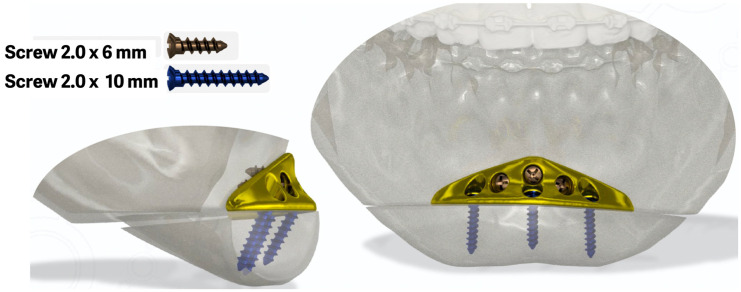
Planning of the new ACP in relation to the dental roots and with the planning of the screws necessary for the installation of the plate.

**Figure 4 cmtr-18-00042-f004:**
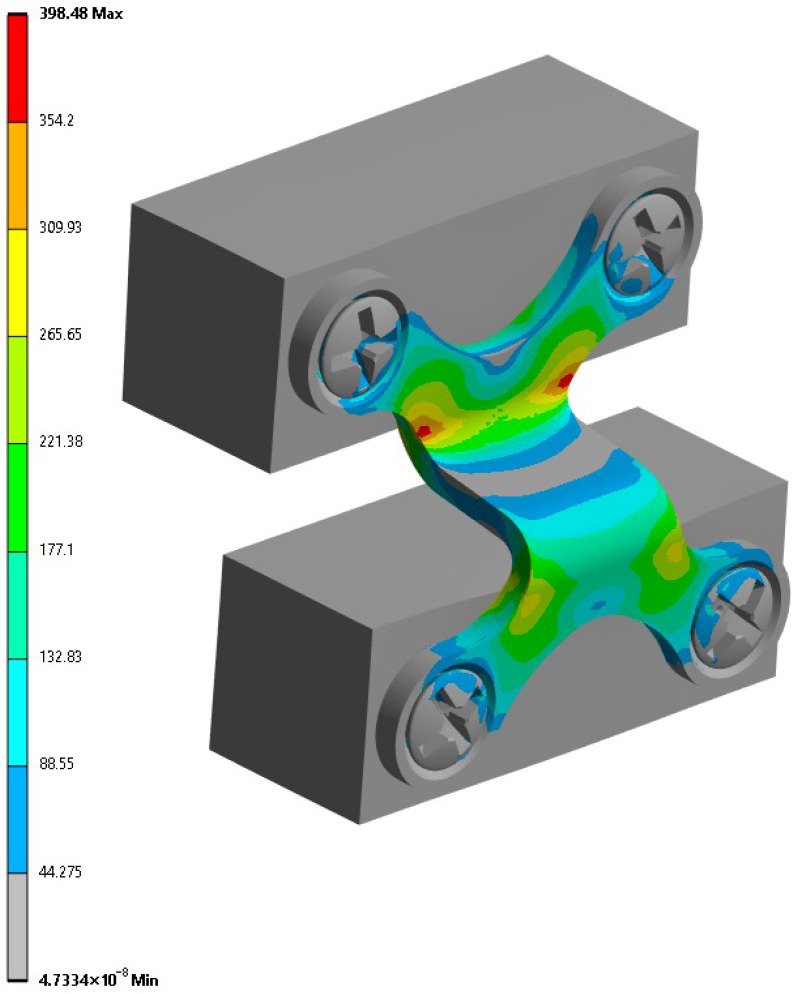
Finite element analysis of the Conventional Plate (CP).

**Figure 5 cmtr-18-00042-f005:**
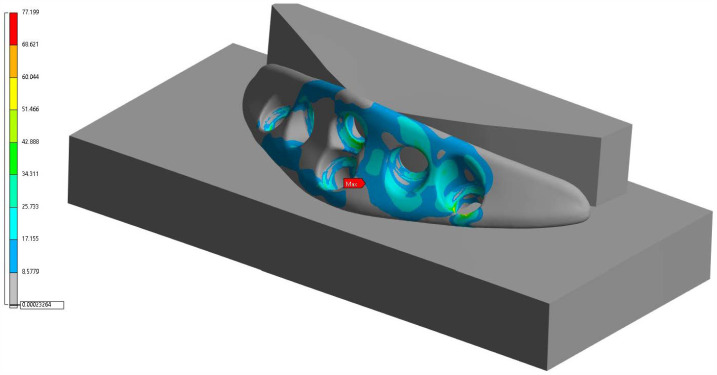
Finite element analysis of the Anatomical Chin Plate (ACP).

**Table 1 cmtr-18-00042-t001:** Comparison between models CP and ACM plates.

Criterion	CP	ACP
	Plate A (Conventional)	B-Plate (Anatomical)
Maximum tension (von Mises)	398.48 MPa	77.19 MPa
Critical Concentration Site	Between the center holes	Around the side holes
Risk of plastic deformation	Moderate	Low
Stress distribution	Concentrated	Homogeneous
Potential for fatigue failure	Bigger	Minor

## Data Availability

The raw data supporting the conclusions of this study are available from the authors upon reasonable request.

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
