# Peer review of "Integrative Innovation in Genioplasty: Advanced 3D Plate Design: Promoting Stability, Aesthetics, and Harmony Excellence"

_1943-3883, 2025, doi:10.3390/cmtr18030042_

Round 1

Reviewer 1 Report

Comments and Suggestions for Authors

This is an interesting and promising research in which a new type of PSO  (Patient Specific Osteosynthesis) is being introduced for fixation in genioplasty. PSO is not new and many advantages for this ACP are general PSO advantages
The plate is innovative and possibly relevant. The manuscript however hampers in methodology and reporting issues and this limits the strength of the conclusions.
- This FEA model has not been validated in vitro or in vivo. Clinical translatability is therefore questionable. Figures of the ACP and the standard osteosynthesis plate suggest a gap in the latter situation and solid contact of the bone parts in the ACP fixation, suggesting possible bias. The ACP plate is thicker than the conventional plate; 1.2 vs 0.8 mm. Diffenrences induced by by plate thickness instead of plate geometry are not taken in account. The applied load is mono vectorial and in contrast to actual muscle forces  on the chinfragment. This limitation is not discussed.

The conclusion that the ACP provides superior biomechanical behavior and improved aesthetic potential. is in my opinion premature. Esthetic outcomes were not presented..

Manuscript contains subjective claims: Should be changed to neutral language.

Authors have developed ACP and need to be complimented, also a possible conflict of interest may be adressed.

Comments on the Quality of English Language

No comment, I am not native English, for that reason I use ChatGPT for formatting grammar and spelling, if not allowed, please erase my review. I had to tick the box of not using AI, because otherwise it would prevent me to submit

Author Response

Dear Reviewer

Thanks for the comments, rest assured that all corrections will be made to the manuscript. T

The initial proposal is to find out if this plate, even though it is more aesthetic, would be viable for future in vivo studies. This study is still at an early stage and our goal is to report the new plaque to the scientific community. Future studies will be carried out comparing several osteotomies for genioplasty, different plate sizes and in vitro biomechanical studies.

I have added some data in finite element analysis, I hope it will make the work more understandable. The force vector at 45º simulates the positioning that all muscles exert on the chin, the analysis using this vector follows the pattern of other studies that did the same finite element analysis in the chin region.

I hope you like it.

Reviewer 2 Report

Comments and Suggestions for Authors

Interesting paper.

Could be improved with actual experimental models to analyse stress-to-failure tests / screw pull out tests.

This paper appears to test only 1 version of the ACP. Should there be other sizes / versions of the ACP, it would be of interest to the readers of similar results can be obtained.

The second comment is with the manufacture of the ACP and CP. I was not able to identify the manufacturer of these devices; if these were created in-house, perhaps the authors should declare this as such.

Author Response

Dear Reviewer

Thanks for the comments, rest assured that all correctionswill be made to the manuscript.

This study is a pilot study and in the future we willperform in vitro biomechanical tests and other studies withdifferent types of osteotomies for genioplasty. I added allthis data to the manuscript.

I hope you like it.

Reviewer 3 Report

Comments and Suggestions for Authors

This article only deals with one type of genioplasty, namely classic genioplasty. Therefore, it is necessary to emphasize it in the article. Such findings may not be true for other types.

Author Response

Dear Reviewer

Thanks for the comments, rest assured that all correctionswill be made to the manuscript.

I will emphasize in the manuscript that classicalgenioplasty was the only one to be used, in the future thiswork intends to explore future osteotomies with differentplate sizes. I related these observations in the discussion.

I hope you like it.

Reviewer 4 Report

Comments and Suggestions for Authors

This is an interesting study evaluating the stress biomechanics of a novel chin titanium plate in comparison to a conventional plate used in genioplasty. These preliminary head-to-head results are encouraging and appear to show superiority of the novel plate, further in vitro and in vivo studies will allow for these results to ultimately be verified. While there are limitations to preliminary studies of this nature, this is well-performed and well-written. 

Author Response

Dear Reviewer

Thanks for the comments, rest assured that all corrections will be made to the manuscript.

I will emphasize in the manuscript that more in vitro and in vitro studies need to be performed. In the future we will test different types of osteotomy and different plate sizes.

Thank you for encouraging our manuscript.

I hope you like it.

Reviewer 5 Report

Comments and Suggestions for Authors

Clarification of the terminologies used would be helpful. For example Blender; a computer graphics software, ANSYS; a computer modelling and simulation software and FEA; a computerised method of analysing the behaviour of structures 

In the conclusion, a clarification that the findings are based on virtual simulations would be helpful. In addition a mention that use of the anatomical chin plates would be limited to centres using virtual surgical planning and cutting guides

Author Response

Dear Reviewer

Thanks for the comments, rest assured that all correctionswill be made to the manuscript.

Terminologies were added to the manuscript and theirconsiderations in the conclusion were added giving more emphasis to the manuscript.

I hope you like it.

Round 2

Reviewer 1 Report

Comments and Suggestions for Authors

Earlier remarks have partially been improved.

Overall the conclusions in terms of excellence (title) and stability are not substantiated by the results, some methods are still hampered

Comments on the Quality of English Language

Needs editing by native english person

Author Response

REVIEWER. >Earlier remarks have partially been improved.Overall the conclusions in terms of excellence (title) and stability are not substantiated by the results, some methods are still hampered.Needs editing by native english person

REPLY:

To Reviewer

Thank you for your prompt reply.

We have changed some parts of the manuscript, mainly regarding the methodology and conclusion.

We would like to emphasize that the ACP plate is still under development and has been used in only 6 clinical cases, cases that have obtained expressive aesthetic results.  To start the validation of this new plate by the scientific community, we chose to do this initial work only with the study of finite elements, thus presenting a new type of plate to the scientific community.

Within our doctoral program, case series studies, biomechanical studies are being programmed and I believe that in the future we will have results with greater robustness.

About the title, we like the part that refers to stability, as it indicates that the ACP plate produces stability over the necessary loads, indicating a fundamental characteristic for the success of the new material.

Native English was used through the EDITAGE software, issuing a certificate of fluency and adequacy to scientific circles. We will attach or certificate.

I hope you like the changes and we are available for any changes.
